# META-LoRA: META-LEARNING LoRA COMPONENTS FOR DOMAIN-AWARE ID PERSONALIZATION

## ABSTRACT

Personalizing text-to-image models to create subject-specific content from limited images is a critical challenge in generative AI. Current methods force a difficult choice between slow, high-fidelity fine-tuning and fast, tuning-free approaches that can struggle with identity details and often replicate the reference pose. We introduce Meta-Low-Rank Adaptation (Meta-LoRA), a novel framework that enhances LoRA-based personalization by meta-learning a domain-specific prior for human identity. Our key insight is to learn a shared, low-dimensional manifold of general identity features from multiple subjects, which provides a powerful foundation for rapidly adapting a small, identity-specific component to a new person from a single image. To enable a rigorous evaluation that addresses pose-copying biases, we introduce Meta-PHD, a diverse benchmark dataset, and R-FaceSim, a robust new similarity metric. On this benchmark, Meta-LoRA achieves a 1.67x faster convergence than standard LoRA while reaching superior identity fidelity. Our findings show that Meta-LoRA not only outperforms its direct baseline but also achieves a more effective balance between identity preservation and prompt adherence than state-of-the-art tuning-free methods. More broadly, our work demonstrates that meta-learning provides a practical and efficient pathway for adapting large generative models, bridging the gap between existing fine-tuning and conditioning-based paradigms. The code, model weights, and dataset will be released publicly upon acceptance.

## 1 INTRODUCTION

Text-to-image generative models have seen significant advancements in recent years, demonstrating a remarkable ability to generate high-quality images from textual prompt encodings (Radford et al., 2021). Among them, latent diffusion models (LDMs) (Rombach et al., 2022; Stability AI, 2024; Labs, 2024; Podell et al., 2024) have proven to be particularly effective, utilizing deep learning to iteratively refine images within a latent space. However, achieving identity personalization (i.e., generating images that accurately capture a specific subject's likeness while maintaining generalization) from a single or few image(s) remains a significant challenge (Zhang et al., 2024).

The mainstream approaches to generative model personalization fall broadly into two extremes. On the one end, methods such as DreamBooth (Ruiz et al., 2023) and textual inversion (Gal et al., 2023) rely on general-purpose fine-tuning algorithms for personalization purposes. While effective in many scenarios, these methods rely purely on the priors embedded into the pre-trained generative model. However, such priors often provide insufficient specialization for capturing subtle domain-specific nuances, such as fine facial details. To overcome such limitations, it is commonly necessary to use a rich set of subject examples and update significant portions of model parameters in order to achieve accurate identity adaptation. Parameter-efficient fine-tuning techniques, e.g., Low-Rank Adaptation (LoRA) (Hu et al., 2022) and its variants (Gao et al., 2024; Borse et al., 2024; Li et al., 2024a; Zhang et al., 2023b; Kopiczko et al., 2024), attempt to reduce adaptation complexity by constraining parameter updates through carefully designed structures; however, as general fine-tuning methods, they share many of the same limitations.

At the other extreme, feed-forward conditioning-based approaches (Ye et al., 2023; Li et al., 2024b; Peng et al., 2024; Guo et al., 2024; Wang et al., 2024; Wei et al., 2023) train ControlNet-like (Zhang et al., 2023a) mechanisms to adapt generative models on-the-fly, circumventing the need for iterative

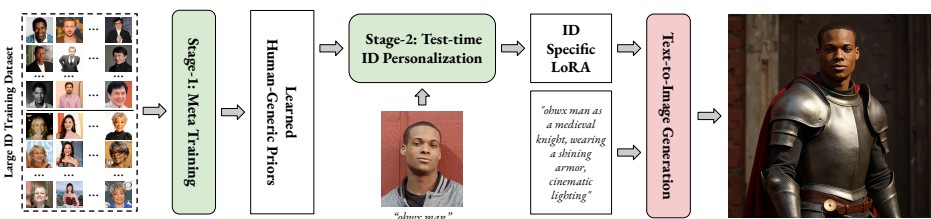

Figure 1: Our overall pipeline. Stage 1 (meta-learning) trains shared LoRA components that capture domain priors. Stage 2 (personalization) fits identity-specific components from a single image.

fine-tuning at test time. Although appealing due to their tuning-free adaptation ability, these methods require large-scale training datasets and complex conditioning networks. In addition, their purely feed-forward structure practically limits their ability to capture fine-grained identity details.

In this work, we aim to combine the strengths of two paradigms: leveraging domain-specific personalization priors while avoiding reliance on computationally expensive feed-forward conditioning modules. To address the limitations discussed above, we propose Meta-Low-Rank Adaptation (Meta-LoRA), a novel two-stage framework that enhances LoRA-based identity personalization while remaining simple and efficient. An overview of the approach is shown in Figure 1. In the first stage, a subset of shared LoRA modules is meta-trained across a large number of identities to capture robust domain-specific priors. In the personalization stage, these shared layers are kept frozen, while compact, identity-specific components are fine-tuned from just a single reference image. This yields a ready-to-adapt LoRA structure that enables efficient and precise identity adaptation without the high-complexity modules required in prior work (Ye et al., 2023; Li et al., 2024b; Peng et al., 2024; Guo et al., 2024; Wang et al., 2024). As a result, this design improves identity preservation compared to both standard LoRA baselines and tuning-free methods. Moreover, by yielding low-complexity LoRA modules, Meta-LoRA accelerates training by 1.67x relative to conventional LoRA and substantially reduces the amount of training data required compared to tuning-free models.

We evaluate the proposed approach with a focus on the FLUX.1-dev model (Labs, 2024), a state-of-the-art text-to-image diffusion system known for its high image quality and strong prompt adherence. Following prior studies, we assess personalized models trained with Meta-LoRA based on (i) the fidelity of generated faces to target identities and (ii) the prompt following abilities. However, our analysis highlights inconsistencies in recent literature regarding dataset choices and evaluation protocols, particularly the problematic reuse of the same image for both personalization and evaluation, which yields unreliable performance estimates. To address these issues, we introduce Meta-PHD, a robust benchmark featuring diverse identities across multiple sources, with varied poses, lighting, and backgrounds, ensuring a rigorous evaluation of identity personalization methods. Our qualitative analysis further complements the quantitative results, capturing subtle yet significant details that conventional metrics may overlook. Together, our results demonstrate that the proposed approach not only preserves identity across diverse scenarios but also maintains strong generalization and prompt adherence, highlighting the potential of meta-learning for enhancing LoRA-based ID personalization.

In summary, our contributions are four-fold: (1) a novel Meta-LoRA architecture that separates shared domain priors from compact, subject-specific components, enabling highly efficient personalization; (2) a practical meta-learning paradigm that learns an optimal manifold for future adaptation without requiring computationally expensive unrolled optimization. This enables a one-shot process that is both **1.67x faster** than standard LoRA and achieves **superior identity fidelity**; (3) a new public benchmark, **Meta-PHD**, curated with diverse poses and identities to address inconsistencies and raise the standard for evaluation in the field; and (4) the **R-FaceSim** metric, an intentionally simple modification to standard evaluation that fills a critical gap by correcting for pose-replication biases, which our user study confirms inflates conventional scores.

## 2 RELATED WORKS

Identity personalization methods can be grouped into two: fine-tuning and feed-forward conditioning-based approaches. In a pioneering work, Ruiz et al. (2023) introduces Dreambooth, a fine-tuning

approach that balances training with a class-specific prior-preservation loss to maintain model generalization. Custom Diffusion (Kumari et al., 2023) fine-tunes select cross-attention parameters for integrating novel concepts. LoRA (Hu et al., 2022) offers a parameter-efficient alternative to fine-tuning parameters directly, enabling controlled adaptation by introducing low-rank updates to the model weights. Given its widespread adoption, we focus on the original LoRA formulation, though Meta-LoRA can naturally extend to other LoRA variants, e.g., Borse et al. (2024); Gao et al. (2024); Li et al. (2024a); Zhang et al. (2023b); Kopiczko et al. (2024).

In the second group of methods, several feed-forward conditioning methods have recently been proposed (see Zhang et al. (2024) for a recent survey). For example, PortraitBooth (Peng et al., 2024), PhotoMaker (Li et al., 2024b), InstantBooth (Shi et al., 2024), PhotoVerse (Chen et al., 2023), FastComposer (Xiao et al., 2024), InstantID (Wang et al., 2024), and PuLID (Guo et al., 2024) all incorporate an IP-Adapter-like design, leveraging identity-related embeddings extracted from input ID images to manipulate text-conditioned generation with various enhancements: PortraitBooth and PhotoMaker fine-tune the diffusion model, with PortraitBooth further introducing an identity loss and locational cross-attention control. InstantBooth integrates an adapter layer positioned after the cross-attention mechanism to inject conditional features into U-Net layers. InstantID employs a ControlNet module that incorporates facial landmarks and identity embeddings. FastComposer incorporates subject embeddings extracted from an image encoder in order to augment the generic text conditioning in diffusion models. In PhotoVerse, a dual-branch conditioning mechanism operates across both text and image domains, while the training is further reinforced by a facial identity loss. PuLID enhances identity preservation through ID loss and improves text-image alignment via an alignment loss. In contrast to these works, Meta-LoRA does not require training a complex conditioning model. In addition, we simply use only diffusion loss; although the incorporation of additional losses is an orthogonal direction, and can easily be incorporated into the Meta-LoRA training, if desired.

As a work mixing elements from both groups of personalization approaches, HyperDreamBooth (Ruiz et al., 2024) adopts a two-stage pipeline: first, a meta-module is trained to generate LoRA-like weights based on an identity image; second, these generated weights undergo fine-tuning for the specific identity. While this method aligns with our work to some extent, it introduces several limitations that hinder its scalability and applicability. As it requires training of a complex image-to-LoRAs mapping network, expressed by a recurrent transformer architecture. In contrast, we propose to simply meta-learn a subset of LoRA components to embed domain-specific priors.

Weights2weights (Dravid et al., 2024) is another related study that constructs a PCA subspace from thousands of fine-tuned models, with personalization via coefficient optimization. In contrast, Meta-LoRA introduces a meta-learned prior through a structured architecture, explicitly minimizing future personalization error. Our method is orthogonal to weights2weights, as Meta-LoRA prescriptively learns an adaptation mechanism for single-image personalization instead of navigating a descriptive weight space.

Finally, we should note that our work has been inspired by the prior work on meta-learning for classification and reinforcement learning tasks. We similarly embrace the meta-learning for fast-adaptation idea pioneered by MAML (Finn et al., 2017). The formulation and technical details of Meta-LoRA, however, are vastly different virtually in all major ways.

## 3 METHODOLOGY

In this section, we provide a detailed explanation of Meta-LoRA, including the model structure and an efficient training algorithm.

**Preliminaries.** In diffusion-based text-to-image (T2I) models, Low-Rank Adaptation (LoRA) (Hu et al., 2022) is primarily integrated into the attention mechanisms of the U-Net backbone, particularly within the cross-attention layers that facilitate interactions between latent image representations and text embeddings. By incorporating low-rank decomposition matrices into the weight update process, LoRA enables efficient fine-tuning while minimizing modifications to the original model parameters. In more advanced architectures, such as Stable Diffusion XL (Podell et al., 2024) and FLUX.1 (Labs, 2024), LoRA may also be applied to transformer-based components, enhancing latent

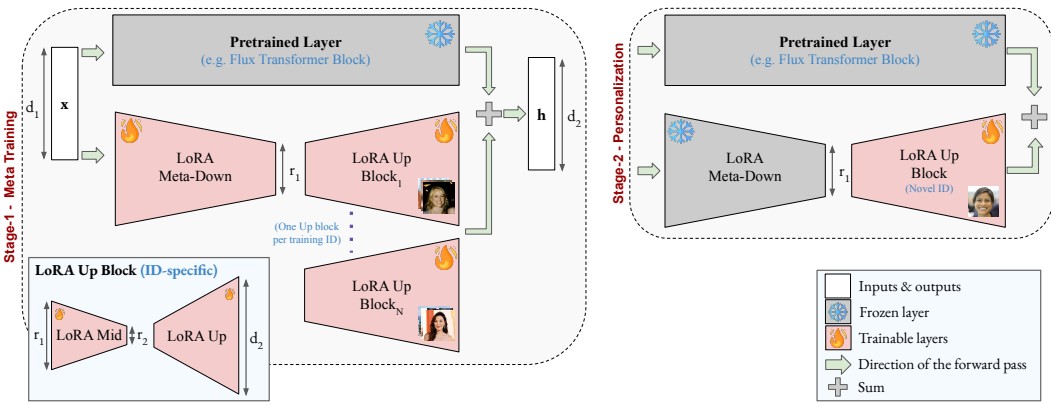

Figure 2: Proposed Meta-LoRA architecture. LoRA Meta-Down components encode identity-independent domain priors, while LoRA Up Blocks are identity-specific. In meta-training (Stage 1), both Meta-Down and temporary Up Blocks for training IDs are optimized. At test time, Meta-Down layers remain frozen, and new Up Blocks are initialized and trained for novel IDs.

feature extraction and improving T2I coherence. By limiting adaptation to these critical layers, LoRA optimizes parameter efficiency while maintaining the integrity of the underlying diffusion model.

We integrate our Meta-LoRA architecture into FLUX.1-dev, a state-of-the-art T2I generation model. However, the Meta-LoRA design follows a generic approach, making it adaptable to other models with similar architectures.

**Meta-LoRA Architecture.** We propose a three-layer adaptation framework by re-defining the original LoRA Down-block as an identity-shared adaptation component, which we call LoRA Meta-Down (LoMD), and decomposing the conventional LoRA Up-block into two sub-components, which we call LoRA Mid (LoM) and LoRA Up (LoU). Specifically, given some pre-trained linear layer $W_0 : \mathcal{R}^{d_1} \to \mathcal{R}^{d_2}$ and its input $x$, Meta-LoRA applies the following residual update:

$$h = W_0 x + \Delta W x = W_0 x + L_{\text{up}}^i L_{\text{mid}}^i L_{\text{meta-down}} x \qquad (1)$$

where $L_{\text{meta-down}}$, $L_{\text{mid}}^i$, and $L_{\text{up}}^i$ correspond to LoMD, LoM and LoU, respectively. The index $i \in \{1, ..., N\}$ indicates the identity among $N$ training identities.

Each LoMD component serves as an identity-independent module shared across all identities to model the personalization manifold for the targeted domain (Figure 2, left). Once trained, LoMD remains frozen during test-time personalization, acting as a learned prior that facilitates rapid adaptation (Figure 2, right). The second and third components, LoM and LoU, are identity-specific, and therefore, parameterized separately for each identity. LoM reduces the output dimension of LoMD from $r_1$ to a lower-rank representation $r_2$, improving efficiency while retaining relevant features, whereas LoU transforms this reduced representation back to the same dimensionality ($d_2$) as the output of the pre-trained layer $W_0$, ensuring compatibility with the base model.

**Training overview.** Our training pipeline comprises two main stages: meta-training and test-time ID personalization. During meta-training, we learn a shared LoRA Meta-Down (LoMD) module for general features while employing distinct LoRA-Mid (LoM) and LoRA-Up (LoU) layers for character-specific details. In the test-time personalization stage, we fine-tune LoM and LoU based on the given identity image. Following subsections provide a detailed explanation of each stage.

**Stage-1: Meta-Training.** The key challenge in the meta-training stage is learning a robust meta-representation in LoMD, while preventing under- or overfitting in the identity-specific LoM and LoU layers. A simple training approach could be optimizing all LoRA components simultaneously, for all characters. However, it is not possible to construct batches large enough to cover all identities. Trying to naively remedy this problem by applying stochastic gradient descent to the LoM and LoU components of only the identities available in each batch (and the shared LoMD parameters) also

yields poor training efficiency, since the identity-specific parameters are updated much less frequently than the shared ones, resulting in poor gradients for LoMD updates.

To achieve efficient training with small batch sizes, we divide the dataset into buckets, each containing a subset of training identities. To minimize the I/O overhead, we make model updates for $q_{\text{bucket}}$-many iterations using each bucket data. We note that buckets can contain data larger than the maximum batch size that the VRAM allows. Importantly, at the beginning of a new bucket, we first apply an adaptive *warm-up procedure*: in the first $q_{\text{warm-up}}$ iterations of a new bucket, only the LoM and LoU layers corresponding to the bucket contents are updated, syncing them with the current status of the LoMD components.

In practice, we set $q_{\text{bucket}}$ such that each example in a bucket is utilized for 10 iterations. We set $q_{\text{warm-up}} = 0.4 \cdot q_{\text{bucket}}$ empirically to ensure that LoMD is updated only after the character-specific layers have sufficiently adapted. In the remaining iterations, LoMD is refined using more informative gradients, ensuring that the learned meta-representation continues to improve without being biased by individual identities. For all model updates, we simply use the latent-space diffusion loss $\mathcal{L}_{\text{diff}}$.

This structured approach prevents poor generalization due to under-trained identity-specific layers, while also mitigating the risk of overfitting. Throughout the meta-training stage, LoMD modules accumulate the common domain knowledge from which all individual-specific personalized models are expressed. At the end of this phase, all trained LoM and LoU components are discarded, and only LoMD components are retained.

**Stage-2: Test-time ID Personalization.** In the second stage, we train only the LoM and LoU layers using a novel target identity image, while keeping the LoMD weights frozen. The $r_1$ dimension (i.e., the LoMD output dimension) remains unchanged, whereas $r_2$ (i.e., the LoM output dimension) is set to empirically determined small values to mitigate overfitting due to the single-image input. To further prevent overfitting on a single image, we apply data augmentations. In total, we generate up to 25 augmented images from a single input. The details of the augmentation process are written in Appendix D.

Although our focus is one-shot personalization, Meta-LoRA naturally extends to settings with multiple reference images. In such cases, the $r_2$ dimension can be increased to enhance model capacity, potentially improving identity fidelity - a promising direction for future exploration. Finally, for simplicity, the final model can be converted back into the default LoRA structure by multiplying the LoMD and LoM matrices, producing a rank-$r_2$ (rank-1 in our experiments) LoRA model. All results presented in this paper are obtained using this conversion.

In Appendix A, we further discuss on the main differences between Meta-LoRA and standard LoRA to elaborate on our novelty.

## 4 EXPERIMENTS

In this section, we evaluate our approach through both quantitative and qualitative analyses, comparing it against state-of-the-art text-to-image personalization methods. We begin by describing the evaluation dataset, experimental setup, and metrics, followed by a presentation of comparative results and corresponding discussions. The main paper focuses on the *'female'* class, as it provides more instances for stage-1 training. Evaluations on *'male'* are included in Appendix I.

### 4.1 BENCHMARK DATA SET

To rigorously assess personalization performance, we curated the Meta-LoRA Personalization of Humans Dataset (**Meta-PHD**), which will be made publicly available to facilitate fair comparisons.

**Meta-PHD-Unsplash.** Its first component is inspired by the Unsplash-50 dataset (Gal et al., 2024) and is sourced from Unsplash under permissive licenses. It comprises 98 images collected from 16 identities (with 4–10 images per identity). Its main novelty is that within each identity, subjects appear in similar attire and backgrounds but exhibit varied poses. This design ensures the dataset captures a range of views, thus preventing models from succeeding by merely replicating a single

reference pose, which is foundational for the proposed R-FaceSim metric. For this component, we designed 10 diverse prompts per identity that enforce a frontal, unobstructed view of the subject.

**Meta-PHD-FFHQ.** The dataset's second component contains 60 high-quality face images (30 male and 30 female) selected from the FFHQ test set (Karras et al., 2019). Instead of utilizing the publicly available Unsplash-50 dataset as in the previous studies (Gal et al., 2024; Guo et al., 2024), we aim to create a more diverse set that covers a broad range of ages and skin tones to ensure broad applicability. We applied strict selection criteria to each image: only one individual is present, the face is unobstructed (i.e., no sunglasses, hats, or hands), the eyes are open, and the head is oriented within 30 degrees of frontal. To evaluate performance, we prepared 20 prompts per gender inspired by prior work (Ruiz et al., 2024; Wang et al., 2024; Ye et al., 2023; Guo et al., 2024; Shi et al., 2024) to test both stylistic transformations and recontextualizations. The complete prompt lists for each component are shared in Appendix M.

## 4.2 EXPERIMENTAL SETUP

**Training dataset.** We utilize a proprietary data set of male and female images with distinct identities from those in the test benchmark and train two separate models. Each individual is represented by 20 high-resolution images ($> 720$p). The dataset includes $1,050$ female and $400$ male subjects. The dataset is diverse in terms of age, skin tone, image environments, and accessories.

**Baseline models.** Although our study is mainly an advancement over the standard LoRA baseline in terms of adaptation time and output quality (i.e., higher and more balanced face similarity and prompt adherence), we additionally provide a comprehensive comparison against three additional state-of-the-art models: InstantID (Wang et al., 2024), PhotoMaker (Li et al., 2024b), and PuLID (Guo et al., 2024). Implementation specifics including backbones are detailed in Appendix F.

**Implementation details.** We trained the Stage-1 Meta-LoRA model for 50,000 iterations, on a single A6000 GPU. For the Stage-2 training, we present our results on several iterations between 250 and 750, and on ranks $\in \{1, 2, 4, 8, 16\}$. For the visual results, we fix the rank as 1 to maintain minimal model complexity and set the iteration count to 375 for Meta-LoRA and 625 for the default LoRA. These iterations are decided empirically to balance facial similarity and prompt following ability. Additional and elaborated implementation details can be found in Appendix B.

## 4.3 EVALUATION METRICS AND R-FACESIM

Following recent studies (Ruiz et al., 2024; Wang et al., 2024; Guo et al., 2024; Shi et al., 2024; Ruiz et al., 2023), we use two key metrics to evaluate performance: *CLIP-T* to measure the prompt following ability and the newly proposed *Robust Face Similarity (R-FaceSim)* to assess the face similarity. As some studies include *CLIP-I* for completeness, we also report *CLIP-I* scores in Appendix H. Furthermore, we provide additional details on all metrics in Appendix G.

The commonly used FaceSim metric aims to evaluate how well the generated images maintain the subject's identity based on the cosine similarity between the reference and generated images. The facial embeddings are obtained using the backbone of a face recognition model. However, we observe two major limitations with it. First, facial recognition methods primarily learn to distinguish individuals, therefore, their embeddings rather tend to lack some of the fine-grained identity details, also observed by Chen et al. (2024); Ruiz et al. (2024); Kung et al. (2024) in various contexts. Second, commonly, the same reference images are used for both personalization and face similarity evaluation. However, in feed-forward personalization models, e.g. InstantID (Wang et al., 2024) and PuLID (Guo et al., 2024), we observe a bias towards generating images with the same pose and/or gaze as the reference images. This phenomenon indicates an undesirable limitation of those models. It also tends to artificially inflate the face-similarity scores, because the face recognition embeddings are sensitive to having the same pose or gaze.

To address the former issue, we provide a detailed discussion of the qualitative results. To address the latter one, we propose a novel face-similarity metric that we call *Robust Face Similarity* (**R-FaceSim**). R-FaceSim is computed by excluding the reference image (used for fine-tuning or output-conditioning); we instead compare each generated image to other real images of the same

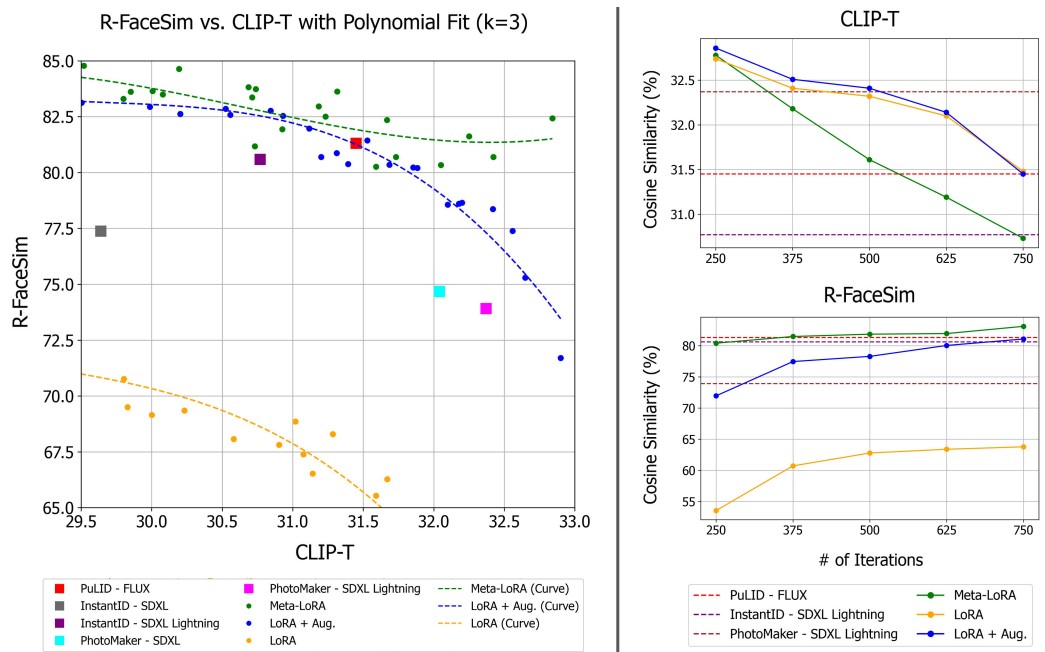

Figure 3: Metric score trends on the *'female'* partition of Meta-PHD. **Left:** R-FaceSim vs. CLIP-T scores for LoRA (Hu et al., 2022) and Meta-LoRA models (ranks ∈ 1, 2, 4, 8, 16) across 250–750 training iterations, with polynomial fit curves for comparison. **Right:** R-FaceSim and CLIP-T over iterations for rank-1 LoRA and Meta-LoRA, compared with PuLID (Guo et al., 2024), InstantID (Wang et al., 2024), and PhotoMaker (Li et al., 2024b). For fairness, PuLID uses the FLUX.1-dev version (Labs, 2024), consistent with Meta-LoRA; InstantID and PhotoMaker are based on SD-XL and SD-XL Lightning (Lin et al., 2024), as their FLUX.1-dev versions are not available.

person (with different poses) and average the cosine similarity. This yields a face-similarity score not inflated by the exact reference image. We encourage future research to adopt this approach to improve the inflated scores from pose copying as acknowledged in Gal et al. (2024).

### 4.4 QUANTITATIVE RESULTS

Figure 3 provides a quantitative comparison of Meta-LoRA against the previously mentioned baselines on only the *'female'* partition of Meta-PHD test set, due to the larger Stage-1 training dataset. The analysis on *male* class is shared in Appendix I. Our evaluation focuses on two key aspects: CLIP-T, which assess prompt adherence; and R-FaceSim, which measures identity preservation.

Base models are typically trained for broad, general-purpose tasks, so fine-tuning them for identity personalization can compromise their prompt following ability, often leading to lower CLIP-T scores. Conversely, overfitting to a reference identity can improve R-FaceSim but at the expense of generalization. Striking a balance between these two objectives remains a challenge for existing identity personalization methods. For example, InstantID (Wang et al., 2024) and PhotoMaker (Li et al., 2024b) tend to favor one over the other: PhotoMaker achieves higher CLIP-T scores but compromises identity preservation, while InstantID emphasizes facial similarity at the cost of broader alignment. In contrast, PuLID (Guo et al., 2024) and our proposed Meta-LoRA demonstrate a more effective balance between identity fidelity and generalization.

On Figure 3 (Left), the training results of *Meta-LoRA* occupy a higher position on the R-FaceSim vs. CLIP-T plot compared to all other models. Its polynomial trendline outperforms the rest, with LoRA+Augmentation (Aug) and PuLID being the closest in performance. While *LoRA+Aug* achieves results comparable to PuLID, the default *LoRA* lags behind all competitors, underscoring the value of data augmentation. Moreover, Meta-LoRA exhibits more stable R-FaceSim values across varying

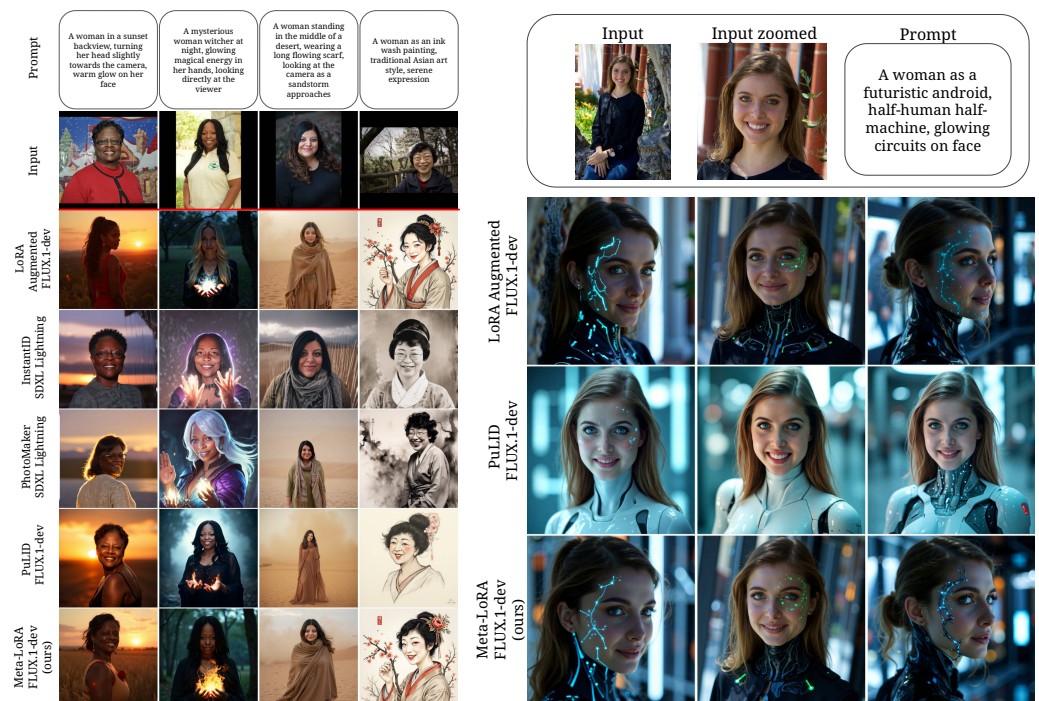

Figure 4: Qualitative comparisons. *Left*: Meta-LoRA preserves ID better while maintaining prompt alignment. Meta-LoRA keeps the pose natural (1st column), promotes identities effectively (2nd and 3rd columns), and follows the given style (4th column). *Right*: Across seeds, Meta-LoRA produces distinct variations while preserving prompt alignment. Details are best viewed on screen, zoomed in.

CLIP-T scores, unlike LoRA+Aug. This suggests that Meta-LoRA trains more consistently and learns facial features faster, as evidenced by higher CLIP-T scores in earlier training stages.

On Figure 3 (Right), the effect of iteration count on CLIP-T and R-FaceSim is shown. The plots reveal an inverse relationship between the two metrics. Meta-LoRA captures facial features more quickly than other LoRA variants, highlighting the efficiency of its meta-learned components in reducing adaptation time. However, this comes at the cost of slightly lower text-image alignment performance when compared at the same iteration. Still, between iterations 375 and 500, Meta-LoRA outperforms PuLID on both metrics, suggesting that this range offers a more optimal balance between identity preservation and text alignment. Additional comparisons are given in Appendix I and L.

Another key difference lies in the scale of pretraining data and the use of additional training losses. Existing models are trained on substantially larger datasets: our dataset constitutes only 0.035% of InstantID's, 1.4% of PuLID's, and 18.75% of PhotoMaker's. Furthermore, PuLID incorporates a face-centric identity loss, while InstantID embeds facial features directly into its network. In contrast, Meta-LoRA achieves comparable or superior performance without relying on such explicit facial losses or features, instead employing a simpler and more efficient training strategy.

To assess perceptual quality beyond automated metrics, we conducted a comprehensive user study. As detailed in Appendix C, the study provides strong empirical evidence for our core contributions: users perceive Meta-LoRA's outputs as being of higher quality and more diverse overall, and our R-FaceSim metric is validated as a necessary correction for pose-overfitting biases.

### 4.5 QUALITATIVE RESULTS

To evaluate visual fidelity and prompt adherence, we qualitatively compare Meta-LoRA against baselines (Guo et al., 2024; Li et al., 2024b; Hu et al., 2022; Wang et al., 2024), focusing on *female* samples. Extra visual results for both genders are shared in Appendix N.

Figure 4 (left) underscores Meta-LoRA's superior ability to preserve nuanced facial characteristics while robustly integrating prompt details. It uniquely achieves a compelling balance between identity retention and scene coherence, producing visually faithful and textually aligned results. In contrast, standard **LoRA** struggles with consistent identity preservation despite following prompts. **InstantID** often replicates the reference image's pose, which, while beneficial for some prompts, significantly curtails its adaptability to diverse viewpoints or styles and can artificially inflate naive similarity metrics. **PhotoMaker** captures general likeness but often fails to render fine-grained facial details, especially when prompts demand significant contextual or stylistic shifts.

Figure 4 (right) examines robustness to seed variation. Meta-LoRA achieves a desirable equilibrium between seed-dependent output diversity and consistent identity fidelity. Unlike standard **LoRA**, which can overfit to incidental background features from the reference image, or **PuLID**, which shows limited variation across seeds and sometimes falters in prompt-driven facial modifications, Meta-LoRA generates diverse yet accurate outputs. This demonstrates its strong capability to explore varied creative outputs while maintaining high fidelity to both the subject's identity and the textual prompt.

In essence, qualitative assessments strongly corroborate Meta-LoRA's quantitative advantages. It consistently produces images that are accurate to the subject's identity and the given prompt, preserving distinctive facial features while offering meaningful creative variation. This positions Meta-LoRA as a highly effective solution for applications demanding both strong identity preservation and versatile prompt adherence.

## 5 CONCLUSION

In this paper, we present Meta-Low-Rank Adaptation (Meta-LoRA), a versatile three-layer LoRA architecture designed to enhance identity personalization in text-to-image generative models. Meta-LoRA employs a two-stage training strategy: the first stage learns generic LoRA Meta-Down layers applicable across multiple identities, while the second stage fine-tunes the LoRA Mid and LoRA Up layers for a specific identity using only a single reference image. By integrating domain-specific priors and optimizing feature learning across diverse identities, Meta-LoRA significantly improves identity preservation, robustness, computational efficiency, and adaptability.

To evaluate our approach, we introduce Meta-PHD, a novel identity personalization dataset used exclusively for testing, and assess Meta-LoRA's performance using this dataset. Our results demonstrate that Meta-LoRA achieves state-of-the-art (SOTA) performance or performs comparably to existing SOTA methods. Moreover, qualitative evaluations highlight Meta-LoRA's ability to generate identity-personalized images while effectively maintaining prompt adherence. Future research directions include the incorporation of recent LoRA variants (Gao et al., 2024; Borse et al., 2024; Li et al., 2024a; Zhang et al., 2023b; Kopiczko et al., 2024) into the Meta-LoRA framework and the exploration of the proposed method's applicability to other model customization tasks.

**Limitations.** Despite significantly reducing personalization training time, Meta-LoRA still requires a dedicated training step per identity, unlike tuning-free methods. While our architecture supports multi-image training for potentially enhanced personalization, its scalability and benefits under such conditions remain unvalidated due to resource constraints; comparisons would also be limited to adaptation-based methods.

Direct comparative evaluation with leading methods like InstantID and PhotoMaker is challenging due to differing base diffusion models (SDXL vs. our FLUX.1-dev). Adapting models for a perfectly fair comparison is resource-intensive. Our primary goal was to show meta-learning's synergy with LoRA for personalization, not to claim universal SOTA status. These methods also use larger, proprietary datasets incompatible with Meta-LoRA's multi-image per-identity meta-training requirement.

Finally, our meta-training dataset has a notable gender imbalance. However, as analyzed in Appendix J, our framework maintains its state-of-the-art performance on the smaller demographic subset, demonstrating its strong data efficiency.

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

# APPENDIX

This document presents additional discussions and details that extend the main paper. The overall structure of the content is outlined below:

## A DETAILED EXPLANATION ON THE DIFFERENCES OF LoRA AND META-LoRA

The key improvement over standard LoRA lies in Meta-LoRA's ability to provide both a more informed initialization and domain-specific regularization during personalization training. The LoRA Meta-Down layers are pre-trained to encode domain-specific priors (e.g., general features of male/female faces) and remain frozen in the second stage.

This allows Stage-2 training to operate on a much smaller number of parameters. For example, if the input dimension is $M$ and output is $N$, Meta-LoRA requires only $128 \times 1 + 1 \times N$ trainable parameters per LoRA block (i.e., LoRA-Down layers are frozen), whereas default LoRA trains $M \times 1 + 1 \times N$, where $M$ is significantly larger than 128. As a result, Meta-LoRA converges about $1.67\times$ faster (375 steps for Meta-LoRA and 625 steps for default LoRA, see Figure 3 in the main paper and in the Appendix Figure 5 for model performance per iteration) and requires fewer resources, fitting comfortably on a 24GB GPU.

Moreover, in addition to its efficiency, Meta-LoRA achieves better identity fidelity and prompt adherence even with fewer training iterations (see Table 4). Qualitative results also support this: in Figure 4 (left) in the main paper, Meta-LoRA better preserves facial features compared to default LoRA with augmentation, and Appendix Figure 8 shows stronger stylistic consistency between FLUX and Meta-LoRA outputs.

While the Stage-2 procedure resembles LoRA by necessity for compatibility, the novelty of our work lies in the meta-training approach, enhancements done to support pre-training across multiple individuals in parallel, and its demonstrated gains in efficiency, scalability, and performance; not in proposing a new LoRA architecture for its own sake.

# B    IMPLEMENTATION DETAILS

## B.1    TECHNICAL DETAILS

Meta-LoRA is implemented using the ai-toolkit[1] framework. Training and evaluation are performed on a single NVIDIA RTX A6000 GPU with 48GB of VRAM (see Appendix F for details). Under this setup, Stage-1 training takes approximately 7 days (13 seconds per iteration for a total of 50,000 iterations), while Stage-2 training takes around 18 minutes (2.8 seconds per iteration for 375 iterations).

For all training runs, we use a learning rate of $0.0004$, set the total number of Stage-1 iterations ($q_{total}$) to 50,000, bucket size ($q_{bucket}$) to 2,500, warm-up iterations ($q_{warm\text{-}up}$) to 1,000, and Stage-2 iterations ($q_{st2}$) to 375. To analyze performance across configurations, we evaluate both Meta-LoRA and the default LoRA at different ranks $\in \{1, 2, 4, 8, 16\}$ and checkpoints $\in \{250, 375, 500, 625, 750\}$. Batch sizes are set to 4 for Stage-1 and 1 for Stage-2. Training is performed using the standard L2 loss, and we use the AdamW optimizer (Loshchilov, 2017).

## B.2    ADAPTATION TO OTHER BASE MODELS

Our method is inherently model-agnostic: it meta-learns an enhanced LoRA structure, and the final trained module can be collapsed into a standard LoRA format. Since standard LoRA fine-tuning has been extensively validated on multiple diffusion backbones (e.g., SD 1.5, SDXL, FLUX), the same compatibility directly applies to Meta-LoRA. The three-layer architecture and multi-identity meta-training strategy are independent of any specific LDM internals, targeting general adaptation mechanisms common across these models. While hardware and dataset-licensing constraints (i.e., allowed to use the stage-1 training dataset on a single A6000 GPU) prevented us from running full Stage-1 meta-training on additional architectures within this submission, we are confident that Meta-LoRA will transfer seamlessly to other LDMs, and we consider this an important avenue for future work.

# C    USER STUDY

We conducted a comprehensive user study with **25 participants**, composed of two parts. Each participant answered **64 questions** presented in a randomized order, for a total of **1600 responses**. The reference images for the questions were randomly and equally sampled from our diverse Meta-PHD-Unsplash and Meta-PHD-FFHQ datasets. The results not only confirm Meta-LoRA's state-of-the-art performance but also provide strong evidence for the necessity of our R-FaceSim metric by highlighting the pose-overfitting issue in existing methods.

## C.1    PART 1 - BEST OVERALL RESULT

Participants chose the single best image from five models (Meta-LoRA, LoRA + aug., PuLID, InstantID, PhotoMaker), answering: *"Considering both the reference person and the text prompt, which is the best overall result?"* We used two evaluation settings to test for pose-overfitting:

- **Multi-Reference (R-FaceSim style):** Participants saw three reference images not used for training, forcing an evaluation of robust identity preservation.
- **Single-Reference (Classic FaceSim style):** Participants saw the single reference image that was used for training.

The results in Table 1 are revealing. The key insight validating our R-FaceSim metric comes from the performance shift between the two settings. The tendency of tuning-free methods to overfit to pose is most evident with InstantID, whose user preference plummets from a high of 27.25% in the Single-Reference setting to just 5.5% in the Multi-Reference one. This drastic drop suggests its high single-reference score is inflated by pose-copying, which is a flaw our R-FaceSim metric is designed to expose. Notably, our Meta-LoRA demonstrates superior generalization, winning not only overall but also achieving the top score in the challenging Single-Reference setting.

---

[1]https://github.com/ostris/ai-toolkit

Table 1: The first part of the user study results. The percentages of the participants' selections are given for InstantID (Wang et al., 2024), PhotoMaker (Li et al., 2024b), PuLID (Guo et al., 2024), Rank-1 LoRA + augmented inputs, and Meta-LoRA. While SD-XL Lightning (Lin et al., 2024) is selected as the base models of InstantID and PhotoMaker, FLUX.1-dev (Labs, 2024) is preferred in PuLID, LoRA, and Meta-LoRA. *Overall* stands for the average of other categories. The greatest results are written in **bold**.

| Model | Base Model | Overall(%) ↑ | Multi-Ref. (%) ↑ | Single-Ref. (%) ↑ |
|---|---|---|---|---|
| InstantID | SD-XL Lightning | 16.375 | 5.500 | 27.250 |
| PhotoMaker | SD-XL Lightning | 6.375 | 5.750 | 7.000 |
| PuLID | FLUX.1-dev | 18.000 | 19.250 | 16.750 |
| Rank-1 LoRA + augmented inputs | FLUX.1-dev | 28.750 | **36.500** | 21.000 |
| Meta-LoRA (ours) | FLUX.1-dev | **30.500** | 33.000 | **28.000** |

## C.2 PART 2 - FIDELITY AND VARIATION

To further investigate model consistency, we asked users to compare the generative diversity of Meta-LoRA against PuLID. These two were selected as they were the top performers in their respective categories (fine-tuning vs. tuning-free) across our prior evaluations. For this part, a single reference image, which is the image that models were fine-tuned/conditioned, was always used. Participants saw three outputs from each model (from different seeds) and were asked: "Considering both diversity and accuracy to the prompt, which model produced the best overall set of images?" This evaluates which model is more reliable across different prompt types (Recontextualized vs. Stylistic).

Table 2: The second part of the user study results. The percentages of the participants' selections are given for PuLID (Guo et al., 2024) and Meta-LoRA. FLUX.1-dev (Labs, 2024) is preferred as the base model in both PuLID and Meta-LoRA. *Overall* stands for the average of other categories. The greatest results are written in **bold**.

| Model | Base Model | Overall(%) ↑ | Recontextualized (%) ↑ | Stylistic (%) ↑ |
|---|---|---|---|---|
| PuLID | FLUX.1-dev | 42.25 | 44.25 | 40.25 |
| Meta-LoRA (ours) | FLUX.1-dev | **57.75** | **55.75** | **59.75** |

As shown in Table 2, users strongly preferred Meta-LoRA's set of images, with a winning margin of over 15 percentage points overall. This preference holds across both prompt categories, demonstrating Meta-LoRA's superior ability to produce consistently high-quality and varied outputs, a feature highly valued by human users.

## C.3 CONCLUSION

This user study empirically validates the principles behind our R-FaceSim metric, demonstrating its alignment with human judgment on pose-overfitting and diversity. Furthermore, it confirms that Meta-LoRA is not only quantitatively superior but also perceptually preferred by a significant margin for its balance of fidelity, prompt-adherence, and creative variation.

## D INPUT IMAGE AUGMENTATION FOR STAGE-2 TRAINING

Here, we share the details of our input image augmentation process before performing Stage-2 personalization training on the input image.

First, we detect the face in the image and determine the longer side length of the facial bounding box ($f_{long}$). Then, we create multiple crops of the image using common aspect ratios: 16:9, 4:3, 1:1, 3:4, and 9:16. For each aspect ratio, we center the face and set the shorter side of the cropped image to one of the following values: $f_{long} \cdot \{1.5, 2, 2.5, 3.5, 4.5\}$. If the original image does not support a specific crop size, we select the closest possible size if an image with that size is not extracted. Additionally, We apply random horizontal flips to each image.

# E Meta-Training and Personalization Algorithms

In this section, we give the algorithms for the proposed meta-training and test-time ID personalization stages. The algorithm for the first stage is shared in Algorithm 1, and for the second stage in Algorithm 2. A detailed explanation of these stages can be found in Section 3.

---

**Algorithm 1:** Stage-1 - Meta Training

---

**Input:**
- $buckets$: the training dataset with the characters are grouped in buckets
- $r_1$, $r_2$: Lora Down and Lora Mid output dimensions - $N$: number of characters per bucket
- $q_{bucket}$: number of iterations with each bucket
- $q_{total}$: number of total iterations
- $q_{warm-up}$: iterations to only update LoM and LoU
- $bs$: batch size

**Output:** Stage-1 weights of Meta-LoRA
$model \leftarrow$ Meta-LoRA.initialize($r_1, r_2, N$) ;
$i_{curr} \leftarrow 0$;
**while** $i_{curr} < q_{total}$ **do**
    **for** $bucket$ in $buckets$ **do**
        $data \leftarrow$ Dataloader($bucket, bs$) ;
        $i_{cb} \leftarrow 0$;
        **for** $batch$ in $data$ and $i_{cb} < q_{bucket}$ **do**
            $loss \leftarrow model(batch)$;
            **if** $i_{cb} < q_{warm-up}$ **then**
                update-mid-and-up-layers($loss$);
            **else**
                update-model($loss$);
            $i_{cb} \leftarrow i_{cb} + 1$;
        $i_{curr} \leftarrow i_{curr} + i_{cb}$;

**return** $model$

---

**Algorithm 2:** Stage-2 - Test-time ID Personalization

---

**Input:**
- $image_{person}$: the single image of the person to learn
- $r_1$, $r_2$: LoRA Down and Mid output dimensions
- $q_{st2}$: Number of iterations
- $weights_{st1}$: Stage-1 LoRA Down weights

**Output:** $weights_{final}$: Final Meta-LoRA Weights
$model \leftarrow$ Meta-LoRA.initialize($r_1, r_2, N$) ;
load-down-weights($model, weights_{st1}$);
$dataset \leftarrow$ augment-and-create-data($image_{person}$);
**for** $iter = 1$ **to** $q_{st2}$ **do**
    **for** $item$ in $dataset$ **do**
        $loss \leftarrow model(batch)$;
        update-mid-and-up-layers($loss$);

**return** $model$

---

# F Baseline Model Details and Implementation Settings

**Model variants and base diffusion backbones.** We evaluate our Meta-LoRA personalization approach against several state-of-the-art subject-driven generation methods with publicly available implementations: InstantID (Wang et al., 2024), PhotoMaker (Li et al., 2024b), and PuLID (Guo et al., 2024). As a FLUX.1-dev-based (Labs, 2024) implementation of PuLID is available, which

is claimed to improve upon the original paper's version (i.e., `FLUX_PuLID_v0.9.1` from its public repository[2]), we use this version for a direct comparison. For InstantID and PhotoMaker, although their respective GitHub demos recommend different diffusion backbones, we standardize the evaluation by running all SDXL-based methods under these two base variants to ensure fairness:

- **SDXL-Base 1.0** (`stable-diffusion-xl-base-1.0`)
- **SDXL-Lightning** (`sdxl-lightning`)

In contrast, our default LoRA and Meta-LoRA approaches are built on a FLUX.1-dev-based (Labs, 2024) model.

**Hardware and data types.**  All experiments based on SDXL are conducted on a single NVIDIA RTX 3090 GPU with 24GB VRAM. In contrast, experiments involving FLUX.1-dev models—including PuLID, LoRA, and Meta-LoRA—are run on a single NVIDIA A6000 GPU with 48GB VRAM, utilizing CPU offloading. Additionally, LoRA training and the stage-2 training of Meta-LoRA (used solely for rank-based performance analysis) are carried out on a single NVIDIA H100 GPU with 66GB VRAM. In all cases, `bf16` is used as the inference data type.

**Model-specific inference configurations.**  For Meta-LoRA and each of the studies that we provide a comparison, the following configurations are selected for inference:

- **Meta-LoRA and LoRA:** `inference_steps=30`, `cfg=3.5`, `true_cfg=1.0`, `lora_strength=1.0`
- **PuLID:** We follow the repository's demo code, and set `id_weight=1.0`, `true_cfg=1.0`, `time_to_start_cfg=1`, `inference_steps=20`, `cfg=4.0`, and `max_sequence_length=128`.
- **PhotoMaker:** We use `PhotoMakerV2`, which differs from the version evaluated in the PuLID paper (Guo et al., 2024) (where its performance was reported as poor, and claimed it is not even compatible with SDXL-Lightning). In our experiments, `PhotoMakerV2` yields substantially improved results—possibly owing to internal updates. To ensure compatibility with SDXL-Lightning (particularly for a 4-step inference), we set `time_spacing="trailing"` in the Euler scheduler configuration and adjust the `start_merge_step` parameter to 0 for SDXL-Lightning or 10 for SDXL-Base. We further prepend the keyword "`img`" to each prompt (e.g., "`man img ...`", "`woman img ...`") to satisfy PhotoMaker's trigger-word requirement.
- **InstantID:** For InstantID, we consider two implementations: **(i)** The `ComfyUI`-based approach used in the PuLID paper (Guo et al., 2024), which exposes a "`denoise`" parameter in the `kSampler` node (with values between 0 and 1). **(ii)** The original InstantID GitHub repository's implementation. In the `ComfyUI` setup, a `denoise` value in the range (0.8, 1.0] yields consistent sigma values ({14.6146, 2.9183, 0.9324, 0.0292, 0.0000} for a 4-step process) and markedly improves image quality for SDXL-Lightning. In the original repository, the time-step derivation departs from the standard SDXL pipeline, resulting in underdeveloped outputs. We therefore replace this part with the official SDXL pipeline (manually applying the aforementioned sigma values) to align the results with those from the `ComfyUI` version. For both implementations, we set `controlnet_conditioning_scale=0.8` and `ip_adapter_scale=0.8`. In spite of employing a robust negative prompt (see below) to counteract SDXL-FLUX artifacts, InstantID outputs tend to exhibit watermarks, a reflection of the training data characteristics. Following the InstantID GitHub recommendations, we generate images at 1016×1016 and then resize them to 1024×1024, which reduces watermark occurrence.

**Common inference configurations.**  Below, the configuration parameters are shared, which are common to multiple models:

- **Image size:** We set the generated image size for all experiments (except InstantID) as 1024×1024.

---

[2] https://github.com/ToTheBeginning/PuLID

- **Negative prompts:** While FLUX.1-dev-based models do not leverage any negative prompts, the following text is used for all SDXL-based methods: *"flaws in the eyes, flaws in the face, flaws, lowres, non-HDRi, low quality, worst quality, artifacts noise, text, watermark, glitch, deformed, mutated, ugly, disfigured, hands, low resolution, partially rendered objects, deformed or partially rendered eyes, deformed, deformed eyeballs, cross-eyed, blurry".*
- **SDXL inference details:** We employ the Euler scheduler for all SDXL inferences. For SDXL-Lightning, we adopt the 4-step `sdxl-lightning` checkpoint from ByteDance's HuggingFace page, with `CFG-scale`=1.2, following recommendations from the PuLID paper (Guo et al., 2024). SDXL-Base 1.0, we set `CFG-scale`=5.0 and use 50 inference steps, in line with standard SDXL configurations.

## G   Details on Evaluation Metrics

**CLIP-T.**   CLIP-T measures how well the generated image aligns with the textual prompt. It is computed as the cosine similarity between the CLIP embeddings of the input prompt and the generated image. We use the CLIP ViT-B/32 model (Radford et al., 2021) to obtain the embeddings. A higher CLIP-T score indicates stronger prompt adherence.

**CLIP-I.**   CLIP-I evaluates image consistency by comparing the output generated with personalization against that generated without personalization. The cosine similarity between these two CLIP embeddings (again computed using the ViT-B/32 variant) reflects the model's ability to inject identity-specific information without distorting the overall scene. Since CLIP-T and CLIP-I measures similer aspects of the model performance and CLIP-I is more dependent on the base model, we mainly focus on CLIP-T for our comparisons and only provide CLIP-I scores for reference in Table 4.

**Conventional FaceSim.**   Conventional FaceSim metric is computed by extracting facial embeddings from the generated images using an Inception-ResNet (Szegedy et al., 2017) model pre-trained on VGGFace2 (Cao et al., 2018), followed by calculating the cosine similarity between these embeddings and those of the real subject. A discussion on the shortcomings of the conventional FaceSim metric, and the proposed improved version is given in Section 4 (i.e. Experiments Section) of the main paper.

**R-FaceSim.**   Algorithm 3 outlines the computation of the Robust Face Similarity Metric (R-FaceSim), which leverages multiple images per identity, carefully curated prompts, and a generative text-to-image model. For each identity, one image is designated as the reference, while the remaining images constitute the test set. The model is either conditioned or fine-tuned on the reference image. For each prompt, the model generates an output image, from which the face is cropped and its identity embedding is extracted. This embedding is then compared—using cosine similarity—to the embeddings of faces from the corresponding test images. The mean similarity for a given prompt and identity is calculated by averaging these cosine similarities. The final R-FaceSim score is obtained by averaging these values across all identities and prompts. The models employed in this process are identical to those used in the *Conventional FaceSim* evaluation.

Table 3: Comparison between the Robust FaceSim and FaceSim metrics, with percentage differences highlighted in cases of significant performance drops. All scores are averaged across both *male* and *female* identity classes.

| Model | Base Model | Face Sim ↑ | Robust Face Sim ↑ | Relative Difference (%) |
|---|---|---|---|---|
| InstantID (Wang et al., 2024) | SD-XL (Podell et al., 2024) | 80.17 | 71.33 | -11.0% |
| | SD-XL Lightning (Lin et al., 2024) | 82.91 | 74.77 | -9.8% |
| PhotoMaker (Li et al., 2024b) | SD-XL | 67.28 | 67.16 | -0.2% |
| | SD-XL Lightning | 66.67 | 63.02 | -5.5% |
| PuLID (Guo et al., 2024) | FLUX.1-dev (Labs, 2024) | 84.76 | 75.72 | -10.7% |
| Rank-1 LoRA (Hu et al., 2022) | FLUX.1-dev | 64.32 | 61.41 | -4.2% |
| + augmented inputs | FLUX.1-dev | 76.77 | 76.15 | -0.8% |
| Meta-LoRA | FLUX.1-dev | 79.45 | 77.16 | -2.9% |

**Justification for R-FaceSim.**   As shown in Table 3, we observe a nontrivial gap between the conventional FaceSim scores and our proposed R-FaceSim scores. In particular, personalization

**Algorithm 3:** Robust Face Similarity Metric (R-FaceSim)

| | | |
|---|---|---|
| | $I$ | Set of identities with multiple images |
| | $P$ | Set of carefully chosen prompts |
| **Input:** | $M$ | Generative text-to-image model |
| | $C$ | Face cropper |
| | $E$ | Identity embedding extractor |

**Output:** $R\text{-}FaceSim$: Robust Face Similarity Score

**for** *each* $i \in I$ **do**
    $ref \leftarrow$ select_one($i$);
    $T \leftarrow i \setminus \{ref\}$;
    $M \leftarrow$ fine-tune/image-prompt($M, ref$);
    **for** *each* $p \in P$ **do**
        $g \leftarrow M(p)$;
        $f_g \leftarrow C(g)$;
        $e_g \leftarrow E(f_g)$;
        $sumsim \leftarrow 0, count \leftarrow 0$;
        **for** *each* $t \in T$ **do**
            $f_t \leftarrow C(t)$;
            $e_t \leftarrow E(f_t)$;
            $sim \leftarrow cosine(e_g, e_t)$;
            $sumsim \leftarrow sumsim + sim$;
            $count \leftarrow count + 1$;
        $S(i,p) \leftarrow \frac{sumsim}{count}$;

$R\text{-}FaceSim \leftarrow mean\{S(i,p) : i \in I, p \in P\}$;
**return** $R\text{-}FaceSim$

approaches such as InstantID (Wang et al., 2024) and PuLID (Guo et al., 2024) often replicate the poses or gazes of the reference image, thereby artificially boosting FaceSim results. Meanwhile, PhotoMaker (Li et al., 2024b) is not as heavily affected by this phenomenon, as indicated by its relatively smaller discrepancy between the two metrics. Additionally, non-augmented LoRA baselines can overfit the position of the face in the generated image, further inflating their face similarity scores. This pose-copying and overfitting phenomenon conflates true identity fidelity with pose or spatial similarity, leading to overestimated accuracy in conventional FaceSim evaluations.

By contrast, R-FaceSim explicitly excludes the original reference image during evaluation and compares each generated output against multiple images of the same identity (captured under diverse conditions). This design ensures that high similarity scores genuinely reflect robust identity preservation rather than a mere replication of pose or gaze. Consequently, the observed differences in Table 3 (ranging up to 10% drop in some cases) highlight how pose-copying and overfitting can lead to inflated identity fidelity under conventional FaceSim. Our R-FaceSim metric thus offers a more stringent and realistic assessment of personalization performance, as it mitigates the bias introduced by reference-image reuse and captures fine-grained identity traits across varied visual contexts. We encourage future work to adopt this approach, especially when evaluating feed-forward or near-instant personalization methods that rely heavily on a single reference image.

## H EXTRA METRIC RESULTS

Table 4 presents a comparison of Meta-LoRA against state-of-the-art models from the literature, evaluated using the CLIP-T, CLIP-I, and R-FaceSim metrics. Meta-LoRA achieves the highest performance in facial similarity (R-FaceSim), establishing a new state-of-the-art in this aspect. For prompt adherence, it ranks second in CLIP-T and third in CLIP-I, following PhotoMaker (Li et al., 2024b). Notably, Meta-LoRA offers a more balanced trade-off between identity preservation and prompt alignment, whereas PhotoMaker tends to prioritize prompt fidelity at the expense of facial similarity.

Table 4: Quantitative comparison of our model with the state-of-the-art publicly available studies in the literature. The scores are taken for five seeds for both *'male'* and *'female'* subsets, and averaged to reach the final output. While CLIP-T is calculated from the Meta-PHD-FFHQ subset, R-FaceSim is computed using the Meta-PHD-Unsplash images. The 1st highest score is highlighted with blue, the 2nd with red, and the 3rd with green.

| Model | Base Model | Train Dataset | CLIP-T (%) ↑ | CLIP-I (%) ↑ | R-FaceSim (%) ↑ |
|---|---|---|---|---|---|
| InstantID (Wang et al., 2024) | SD-XL (Podell et al., 2024)
SD-XL Lightning (Lin et al., 2024) | 60M | 29.37
30.52 | 65.17
70.79 | 72.09
75.26 |
| PhotoMaker (Li et al., 2024b) | SD-XL
SD-XL Lightning | 112K | 31.51
31.92 | 73.55
78.28 | 67.57
63.51 |
| PuLID (Guo et al., 2024) | FLUX.1-dev (Labs, 2024) | 1.5M | 30.95 | 74.66 | 75.72 |
| Rank-1 LoRA
+ augmented inputs | FLUX.1-dev | 0 | 31.63
31.47 | 79.48
77.21 | 61.41
76.15 |
| Meta-LoRA (ours) | FLUX.1-dev | 8K (male), 21K (female) | 31.66 | 77.96 | 77.16 |

# I  ANALYSIS ON THE *Male* CLASS PERFORMANCE

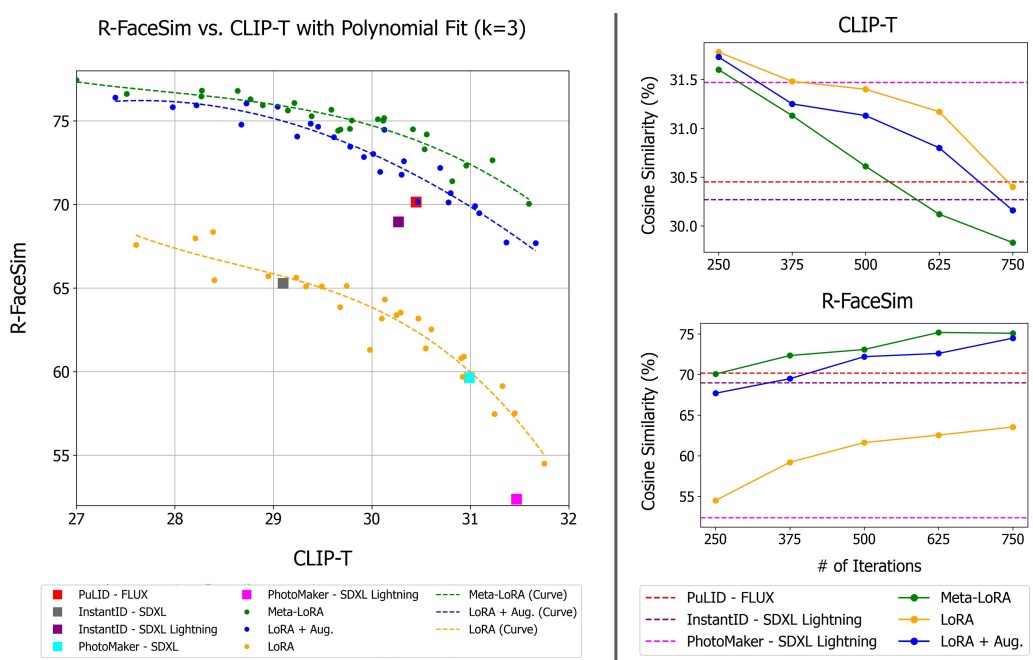

Figure 5: Illustration of the metric score trends for Meta-LoRA and the standard LoRA (Hu et al., 2022) models on the Meta-PHD dataset, focusing on the *'male'* class. **Left:** A plot of R-FaceSim versus CLIP-T scores for all LoRA and Meta-LoRA models trained with ranks ∈ 1, 2, 4, 8, 16 and across 250, 375, 500, 625, and 750 training iterations. The polynomial fit curves per each model are also shared for better comparison. **Right:** R-FaceSim and CLIP-T scores plotted against iteration count for rank-1 LoRA and Meta-LoRA models. These results are compared with state-of-the-art models from the literature, including PuLID (Guo et al., 2024), InstantID (Wang et al., 2024), and PhotoMaker (Li et al., 2024b). To ensure a fair comparison, we use the FLUX.1-dev version of PuLID (Labs, 2024), which is consistent with the Meta-LoRA training setup. InstantID and PhotoMaker are based on SD-XL (Podell et al., 2024) and SD-XL Lightning (Lin et al., 2024), as public implementations of these models are not currently available for FLUX.1-dev.

The left panel of Figure 5 illustrates the distribution of R-FaceSim and CLIP-T scores for the *'male'* class across various hyperparameter configurations. Meta-LoRA consistently outperforms all baseline and previous state-of-the-art models, maintaining higher median and upper-bound scores. Compared to the female class, boundaries between different architectures (i.e., the performance gap) is easier

to observe. Interestingly, even the baseline LoRA model demonstrates improved facial similarity compared to PuLID, suggesting that simpler adaptation methods can be competitive under certain conditions.

In Figure 5 (right), the plots show how CLIP-T and R-FaceSim scores vary with training iteration count. While the trends are similar to those observed for the *female* class (refer to Section 4.4 of the main paper), Meta-LoRA achieves a greater margin in R-FaceSim over prior state-of-the-art models when evaluated on the male subset.

## J    ANALYSIS ON DATASET GENDER IMBALANCE

Our meta-training dataset has a significant gender imbalance, containing images from **1,050 female** subjects compared to only **400 male** subjects. A potential concern with such a data disparity is that the framework's performance might be substantially weaker on the underrepresented demographic.

However, our stratified evaluations demonstrate that Meta-LoRA maintains its state-of-the-art performance across both subsets. As detailed in the analysis in Appendix I, our framework still outperforms all baselines on the male test set, mirroring the state-of-the-art results observed in the female-to-female comparisons.

This robust performance, despite the male-specific training set being considerably smaller, highlights the remarkable data efficiency of our meta-learning paradigm. It shows that Meta-LoRA can establish an effective, task-aware identity manifold from a more constrained number of subjects. This finding is particularly relevant when considering that the large-scale, proprietary datasets of competitor models may also contain demographic imbalances. The ability of our framework to deliver top-tier results from a smaller, transparently-sized dataset underscores its resilience and efficiency. While prioritizing more balanced datasets in future work remains an important goal, these results confirm the framework's effectiveness in data-limited scenarios.

## K    CHOICES OF RANK AND ITERATION COUNT ON META-LORA AND LORA TRAININGS

**On the Stage-1 rank ($r_1$) selection.**    All of our experiments on different $r1$ configurations are conducted under the single-image Stage-2 fine-tuning setting. Due to hardware limitations during Stage-1 training (i.e., permission to use only a single A6000 GPU because of the dataset license issue), we focus on the *female* partition of the dataset and explore $r_1 \in \{64, 128, 256\}$ with a fixed $r_2 = 1$. We track CLIP-T and R-FaceSim metrics over 60,000 Stage-1 iterations (evaluated every 10,000 steps) and 1,000 Stage-2 iterations (evaluated every 125 steps). Our preliminary exploration provided several insights that guided our final parameter selection:

- Increasing $r_1$ leads to faster overfitting and reduced CLIP-T scores in early Stage-2 iterations. In contrast, $r_1 = 64$ results in more stable training curves compared to $r_1 = 256$.
- Across all $r_1$ values, the best balance of CLIP-T and R-FaceSim occurs between 40,000 – 50,000 Stage-1 iterations and 375 – 500 Stage-2 iterations.
- Overall, the most favorable trade-off between identity fidelity and prompt adherence is observed at $r_1 = 128$, Stage-1 iteration 50,000, and Stage-2 iteration 375.

**On the Stage-2 rank ($r_2$) selection.**    Although we present results across various rank values in Figure 3 (Left) of the main paper, all of our qualitative and additional quantitative analyses are conducted using rank-1 configurations. This decision is motivated by several considerations:

- Our goal is to maintain a minimal and straightforward setup, while still demonstrating that the model can effectively learn identity representations. To align with this objective, we also restrict fine-tuning to a single identity image.
- With only one image available during fine-tuning, larger model capacities (i.e., higher ranks) increase the risk of overfitting. Rank-1 offers a good trade-off between CLIP-T and R-FaceSim performance, making it a practical and effective default choice for our study. See Appendix L for a more detailed analysis.

That said, our framework naturally supports scenarios involving multiple identity images during fine-tuning, which can be further explored in future work. In such cases, using higher ranks may further improve and stabilize model performance.

**On the number of iterations.** As discussed in Appendix B, we evaluate our models across different rank values and training iterations. For Meta-LoRA, we empirically observed that iteration 375 achieves a well-balanced trade-off between prompt adherence and identity preservation—that is, the generated images accurately reflect the target identity without exhibiting noticeable overfitting to the input image. Thanks to its Stage-1 preconditioning on identity-related information, Meta-LoRA converges more rapidly than the default LoRA architecture. For example, the default LoRA model reaches a similar balance around iteration 625 (i.e., identity fidelity remains underfitted at earlier iterations). Based on these findings, we select iteration 375 for Meta-LoRA Stage-2 and iteration 625 for LoRA in our main evaluations.

Although these extended and augmented LoRAs show improved performance, our Meta-LoRA with its meta-learned prior, converges faster and achieves higher identity fidelity in fewer steps.

## L  PERFORMANCE CURVES ON DIFFERENT RANKS

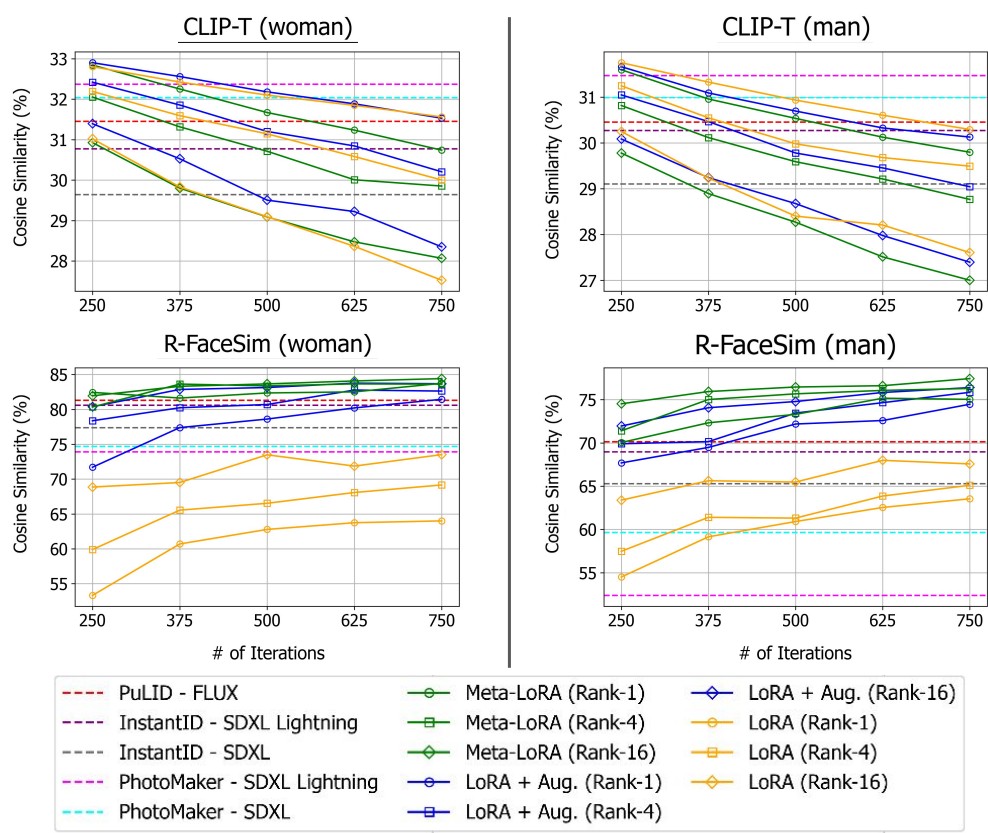

Figure 6: Visualization of CLIP-T and R-FaceSim performance for Meta-LoRA and baseline models across different fine-tuning iteration counts. The plots include results for both gender classes and three rank settings (i.e., 1, 4, and 16).

Figure 6 analyzes the evolution of CLIP-T and R-FaceSim scores across different fine-tuning iterations (i.e., 250, 375, 500, 625, 750) and selected rank values (i.e., 1, 4, and 16). For both gender classes, increasing the rank generally leads to improved facial similarity (higher R-FaceSim) but at the cost of

reduced prompt adherence (lower CLIP-T), indicating a tendency toward overfitting at earlier stages of training.

While CLIP-T scores show a noticeable gap between different ranks across all iteration counts, the variation in R-FaceSim narrows as the number of iterations increases for both Meta-LoRA and LoRA+aug. These trends suggest that lower ranks are better suited for achieving a balanced trade-off between identity preservation and prompt following, particularly in the context of final model evaluations.

# M  DETAILS ON EVALUATION DATASETS AND PROMPTS

The FFHQ partition of the Meta-PHD dataset, along with the prompts listed in Tables 5 and 6, is used exclusively to assess the models' prompt-following capabilities, as measured by the CLIP-T metric. These prompts include both stylistic and recontextualization examples, enabling evaluation across multiple dimensions of prompt adherence.

In contrast, the Unsplash partition and the prompts provided in Table 7 are employed to evaluate our proposed metric, R-FaceSim. These prompts are specifically crafted to encourage the generation of images in which the subject is directly facing the camera, with no facial occlusions, ensuring optimal conditions for facial similarity assessment.

Table 5: Meta-PHD-FFHQ Recontextualized Prompts

| Prompt |
| --- |
| A man in a firefighter uniform, standing in front of a firetruck, looking straight ahead. |
| A man in a spacesuit, floating in space with Earth in the background, face clearly visible. |
| A 90-year-old man with completely white hair, deep wrinkles covering his forehead, around his eyes, and mouth, looking at the viewer, well-lit portrait. |
| A man as a 1-year-old baby, joyful and playful, looking directly at the camera, soft lighting. |
| A man riding a bicycle under the Eiffel Tower, smiling at the camera. |
| A man playing a guitar in the forest, looking at the viewer, natural lighting. |
| A man working on a laptop in a modern office, facing forward with focused expression. |
| A man as a medieval knight, wearing a shining armor, cinematic lighting. |
| A man in a 1920s noir setting, wearing a fedora, moody lighting. |
| A man standing in the middle of a desert, wearing a long flowing scarf, looking at the camera as a sandstorm approaches. |
| A woman in a swimming pool, water reflecting on her face, looking straight ahead. |
| A woman in a sunset backview, turning her head slightly towards the camera, warm glow on her face. |
| A woman piloting a fighter jet, cockpit reflections visible, face clearly in focus. |
| A woman in New York, standing in Times Square, bright city lights illuminating her face. |
| A woman reading books in the library, facing forward, glasses on, scholarly expression. |
| A woman driving a car, cityscape in the background, focused expression, looking ahead. |
| A woman standing in the middle of a desert, wearing a long flowing scarf, looking at the camera as a sandstorm approaches. |
| A woman walking along a misty cobblestone street in an old European town, holding a vintage lantern, gazing into the distance. |
| A woman sitting on a rooftop at dusk, overlooking a glowing city skyline, sipping from a warm cup, smiling at the camera. |
| A woman in a 1920s noir setting, wearing a vintage dress, moody lighting. |

Table 6: Meta-PHD-FFHQ Stylistic Prompts Examples

| Prompt |
| --- |
| A man as a cyberpunk character, neon lights reflecting on his face, highly detailed. |
| A man as an ink wash painting, traditional Asian art style, serene expression. |
| A man as a futuristic android, half-human half-machine, glowing circuits on face. |
| A man in an art nouveau portrait, elegant and highly stylized. |
| A Pixar character of a man, expressive and exaggerated facial features. |
| A man sculpted out of marble, photorealistic, museum lighting. |
| A pop-art style portrait of a man, bright colors and bold outlines. |
| A Funko pop figure of a man, oversized head, cartoonish features. |
| A pencil drawing of a man, highly detailed, face in full view. |
| A painting of a man in Van Gogh style, expressive brush strokes, looking at the viewer. |
| A woman as a cyberpunk character, neon lights reflecting on her face, highly detailed. |
| A woman as an ink wash painting, traditional Asian art style, serene expression. |
| A woman as a futuristic android, half-human half-machine, glowing circuits on face. |
| A woman in an art nouveau portrait, elegant and highly stylized. |
| A painting of a woman in Van Gogh style, expressive brush strokes, looking at the viewer. |
| A Pixar character of a woman, expressive and exaggerated facial features. |
| A woman sculpted out of marble, photorealistic, museum lighting. |
| A pop-art style portrait of a woman, bright colors and bold outlines. |
| A Funko pop figure of a woman, oversized head, cartoonish features. |
| A mysterious woman witcher at night, glowing magical energy in her hands, looking directly at the viewer. |

Table 7: Meta-PHD-Unsplash Prompts - FaceSim

| Prompt |
| --- |
| A [man/woman] standing in a sunlit urban plaza, facing directly toward the camera with a fully visible, highly detailed face and direct eye contact. |
| A [man/woman] in a casual outfit, positioned in a blooming garden with soft natural light, [his/her] face clearly visible and [his/her] eyes meeting the camera. |
| A [man/woman] in a minimalist, modern outfit, set against a softly lit abstract background, with [his/her] face rendered in intricate detail and direct eye contact. |
| A [man/woman] in elegant attire, standing against a scenic landscape at sunrise, [his/her] face captured in sharp detail with a steady, direct gaze. |
| A [man/woman] in a relaxed, stylish ensemble, posed on a quiet beach at sunset, with a fully visible, richly detailed face and clear eye contact. |
| A [man/woman] in a fashionable outfit, placed in a vibrant urban art scene, with [his/her] face prominently visible, detailed, and looking straight at the camera. |
| A [man/woman] with a confident expression, standing in a serene park under bright daylight, [his/her] face rendered in high detail and engaging directly with the viewer. |
| A [man/woman] in a chic outfit, situated in a picturesque alleyway with soft ambient lighting, with [his/her] face clearly displayed in full detail and direct gaze. |
| A [man/woman] in a modern, casual look, standing in a contemporary indoor space with artistic lighting, [his/her] face fully visible and rendered with intricate detail as [he/she] looks at the camera. |
| A [man/woman] in a trendy outfit, positioned in a lively outdoor market where natural light highlights [his/her] features, [his/her] face completely unobstructed and detailed, with direct eye contact. |

## N    QUALITATIVE RESULTS

Figures 7 and 8 demonstrate extended comparison for a diverse set of inputs and reference images.

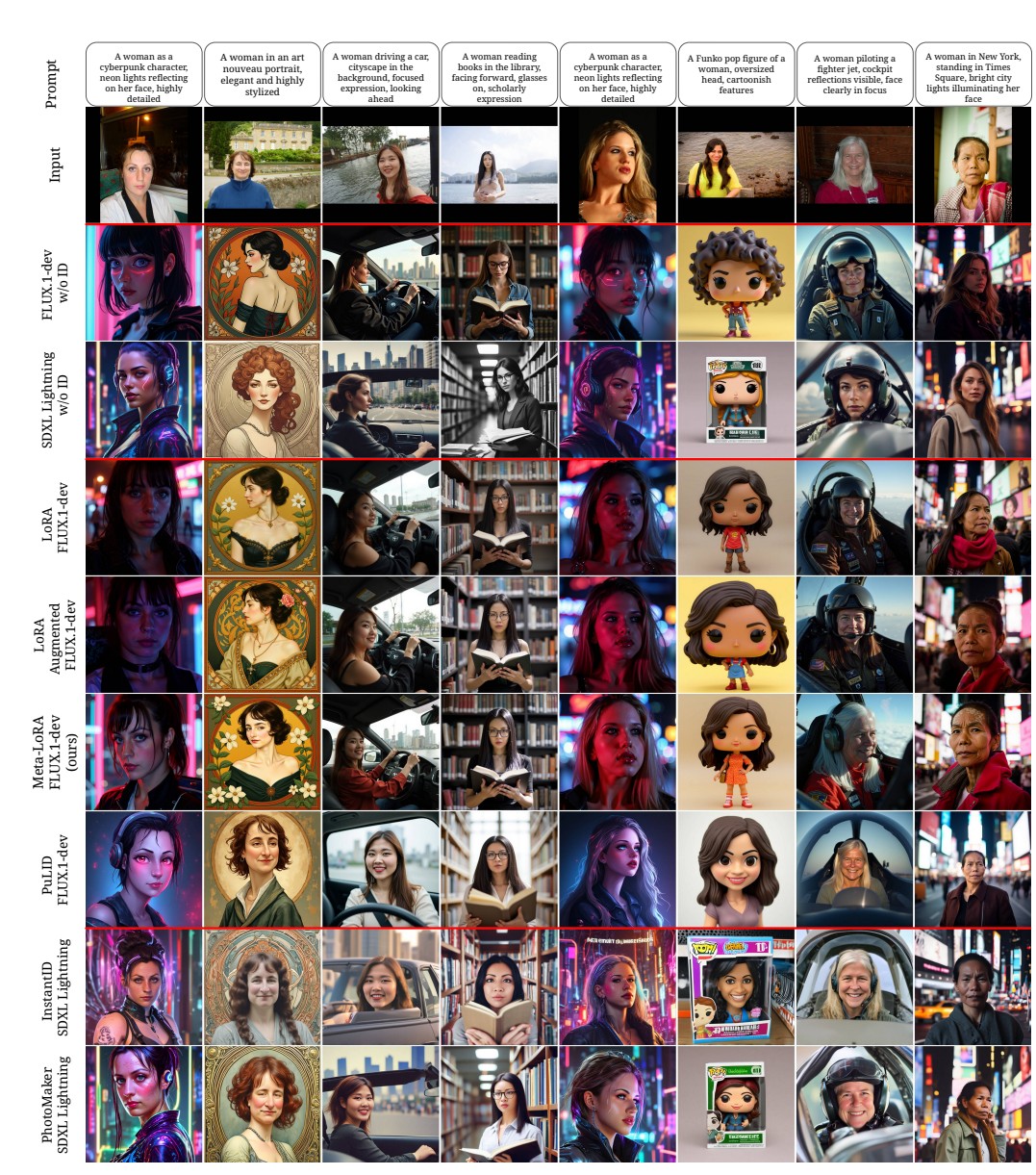

Figure 7: Qualitative comparisons for an extended list of baselines for woman identities.

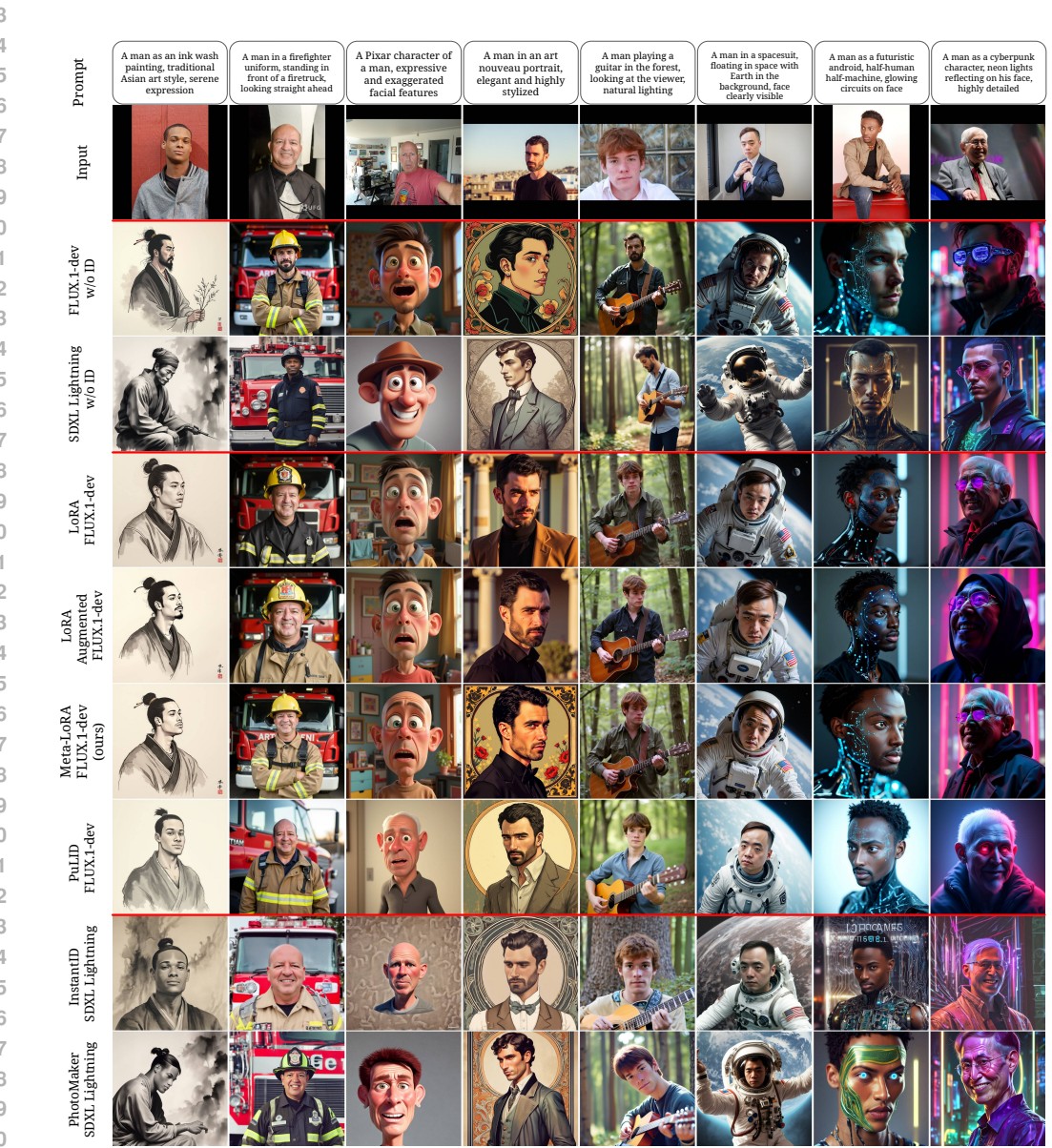

Figure 8: Qualitative comparisons for an extended list of baselines for man identities.

