# OpenReview forum: "Meta-LoRA: Meta-Learning LoRA Components for Domain-Aware ID Personalization"
_ICLR.cc/2026/Conference — ICLR 2026 Conference Withdrawn Submission_

### Official Review · Reviewer_av9Y · 2025-10-26

**Soundness:** 2
**Presentation:** 2
**Contribution:** 2
**Rating:** 2
**Confidence:** 4

**Summary:**

This work introduces Meta-LoRA, a fine-tuning-based approach designed to enhance the fidelity of personalized text-to-image models. By employing a two-stage training paradigm, Meta-LoRA separates shared domain priors from compact, subject-specific components. It accelerates adaptation by 1.67x compared to the standard LoRA.

**Strengths:**

- The paper is well-presented and easy to follow. The methodology is described with sufficient clarity, and the figures effectively illustrate the proposed architecture.
- The decomposition-based LoRA design, separating shared and identity-specific parameter, is a interesting architectural choice that offers a degree of modularity.

**Weaknesses:**

- The proposed method requires both large-scale pre-training and per-identity fine-tuning, inheriting the drawbacks of both paradigms: high computational cost, extensive data requirements, and complex deployment logistics.
- The experimental setup is flawed, as the baselines used for comparison are predominantly zero-shot or non-fine-tuned methods. This comparison is inherently unfair, as the proposed method leverages identity-specific training data while the baselines do not. A more rigorous evaluation should include strong fine-tuning-based competitors.
- The paper lacks discussion of overall efficiency, both during training and, critically, at inference time.

**Questions:**

- In what real-world scenarios does the authors envision this pre-training + ID-specific tuning pipeline being preferable to existing single-stage personalization methods?
- How does the proposed method compare to baselines in terms of computational cost during inference?

---

### Official Review · Reviewer_zbHs · 2025-10-28

**Soundness:** 2
**Presentation:** 2
**Contribution:** 2
**Rating:** 2
**Confidence:** 4

**Summary:**

In this paper, they propose Meta-LoRA, a new framework for the personalization training of text-to-image models. Unlike existing LoRA method, they separate the LoRA parameters into a shared up-LoRA and an identity-specific down-LoRA, designed for training on domain-specific datasets, such as face. This approach achieves faster convergence and a better quality-identity trade off with less training data compared to standard LoRA training. Furthermore, they propose a controlled metric, R-FaceSim, considering that the existing FaceSim metric fails to capture fine-grained identity differences and suffers from identity leakage between training and inference. Overall, the proposed method demonstrates superior trade-offs between quality and identity preservation compared to various existing LoRA training methods and pre-trained models.

**Strengths:**

**S1**. The paper is well-structured and clearly presented, with a solid experimental evaluation. The paper is well-organized and easy to follow.

**S2**. The proposed method demonstrates strong quantitative and qualitative performance, and achieves faster test-time adaptation speed compared to existing approaches.

**S3**. The proposed metric and dataset are well-motivated and sound, and they have the potential to make a valuable contribution to the research community.

**Weaknesses:**

**W1**. The proposed method still requires a curated face dataset and a large number of test-time adaptation iterations, which limits its practical applicability.

**W2**. (This is my major concern) This shared-LoRA concept, in a broader sense, differs primarily in terms of architecture but is conceptually similar to encoder-based methods (InstantID, IP-adapter...). Therefore, a direct comparison with encoder-based approaches incorporating test-time adaptation (e.g., LoRA) is essential.

**W3**. In recent research, human personalization tasks have been extended to handle multiple identities in a single generated image. It is important to include experiments on multi-human scenarios or LoRA merging techniques to evaluate the method’s scalability and generalization.

If my concerns are addressed, I would be happy to reconsider the score.

**Questions:**

**Q1**. Similar to W2, are there any results showing the training speed or metric performance when attaching LoRA to an encoder-based method? Such a setting would likely enable a fairer comparison between the approaches.

**Q2**. Have the authors evaluated the method on non-human domains? While the method itself does not appear to be explicitly face-specific, it would be valuable to examine whether it generalizes to other domains or can be applied to multiple domains simultaneously.

**Details Of Ethics Concerns:**

No concern.

---

### Official Review · Reviewer_tXE3 · 2025-10-31

**Soundness:** 2
**Presentation:** 2
**Contribution:** 2
**Rating:** 2
**Confidence:** 5

**Summary:**

To address the trade-off issue between "slow but high-fidelity fine-tuning" and "fast but low-quality identity detail preservation with a high tendency to replicate reference poses" in the personalization of text-to-image models, this paper proposes the Meta-LoRA framework. This framework meta-learns domain-specific priors for human identity: first, it learns a shared, low-dimensional manifold of general identity features from multiple subjects, and then rapidly adapts identity-specific LoRA Mid (LoM) and LoRA Up (LoU) components from a single image. Experiments have demonstrated the effectiveness of this method.

**Strengths:**

1. This work establishes a rigorous and reliable evaluation system to solve inconsistencies in existing assessments.
2. This work designs an efficient two-stage training paradigm with strong parameter efficiency.
3. This proposed metdho demonstrate data efficiency, comparing to state-of-the-art methods, Meta-LoRA’s meta-training dataset is only 0.035%–18.75% the size of baselines.
4. This paper is well-written and easy to follow.

**Weaknesses:**

1. Meta-LoRA still requires a dedicated training step for each identity, unlike tuning-free feed-forward conditioning methods that avoid test-time fine-tuning entirely.
2. Lack of novelty.
3. The meta-training dataset suffers from significant gender imbalance, with 1,050 female subjects compared to only 400 male subjects, such an imbalance may introduce implicit biases in the learned domain priors and limits the method’s representativeness across diverse groups.
4. Direct comparative evaluation with leading methods (e.g., InstantID, PhotoMaker) is compromised by inconsistent base diffusion models. Meta-LoRA is built on FLUX.1-dev, while InstantID and PhotoMaker use SD-XL or SD-XL Lightning.

**Questions:**

1. Meta-LoRA still requires a dedicated training step for each identity, unlike tuning-free feed-forward conditioning methods that avoid test-time fine-tuning entirely.
2. Lack of novelty.
3. The meta-training dataset suffers from significant gender imbalance, with 1,050 female subjects compared to only 400 male subjects, such an imbalance may introduce implicit biases in the learned domain priors and limits the method’s representativeness across diverse groups.
4. Direct comparative evaluation with leading methods (e.g., InstantID, PhotoMaker) is compromised by inconsistent base diffusion models. Meta-LoRA is built on FLUX.1-dev, while InstantID and PhotoMaker use SD-XL or SD-XL Lightning.

---

### Official Review · Reviewer_pYkt · 2025-11-02

**Soundness:** 3
**Presentation:** 3
**Contribution:** 3
**Rating:** 6
**Confidence:** 4

**Summary:**

The paper proposes Meta-LoRA for personalized text-to-image diffusion models.  Unlike standard LoRA-based methods (e.g., DreamBooth) that learns a LoRA for each subject, Meta-LoRA consists of LoRA Meta-Down for identity-independent domain priors and LoRA Up Blocks for identity-specific features. Meta-LoRA first meta-trains shared “Meta-Down” components capturing domain priors (e.g., general facial features), and then personalizes only small “LoRA Mid” and “LoRA Up” modules from a single reference image. This yields faster convergence (1.67× speedup), improved identity fidelity, and efficient adaptation.
Additionally, this paper introduces a new benchmark for personalized image generation, called Meta-LoRA Personalization of Humans Dataset (Meta-PHD). Moreover, this paper points that current Face similarity metric based on face embedding lacks the evaluation of fine-grained details and contains bias towards generating images with the same pose and gaze. To this end, this paper proposes Robust Face similarity (R-FaceSim), that calculates the similarity between generated images with other images of the same person, rather than the input reference image. Experimental results suggests the proposed method outperforms baseline models.

**Strengths:**

1. The proposed two-stage meta learning method is novel and interesting, as it decouple general facial features and identity-specific feature. Additionally, it has better efficiency with faster convergence speed.

2. The new personalized image generation benchmark is helpful for the community.

3. The concern of current face similarity calculation is reasonable, and the proposed solution (R-FaceSim) seems a practical solution.

**Weaknesses:**

1. Given that a new benchmark is proposed, more evaluation on other methods are preferred. Ideally, this paper could presents a leaderboard for all types of methods. The current submission only evaluates a few methods.

2. This paper only evaluates Meta-LoRA on Flux.1-dev model. The generalization ability of the proposed Meta-LoRA is in doubt. It would be better if this paper could demonstrate that Meta-LoRA is still effective over the standard LoRA on other foundation models, such as SDXL.

**Questions:**

This paper proposes a novel method and a new benchmark for personalized image generation. To further improve this paper:

For the method side, this paper needs demonstrate its effectiveness on other foundation models.

For the benchmark side, this paper needs to present more results of other methods.

---

### Note · Authors · 2025-11-21

I have read and agree with the venue's withdrawal policy on behalf of myself and my co-authors.